

# On the global geodynamic consequences of different phase boundary morphologies

Gwynfor T. Morgan[1], J. Huw Davies[1], Robert Myhill[2], and James Panton[1]

[1]School of Earth and Environmental Sciences, Cardiff University, Park Place, Cardiff, Wales
[2]School of Earth Sciences, University of Bristol, Bristol, England

**Correspondence:** Gwynfor T. Morgan (morgangt2@cardiff.ac.uk)

**Abstract.** Phase transitions can influence mantle convection patterns, inhibiting or promoting vertical flow. One such transition is the ringwoodite-to-bridgmanite plus periclase transition, which has a negative Clapeyron slope and therefore reduces mantle flow between the upper and lower mantle. Interactions between different transitions and significant Clapeyron slope curvature can potentially result in complexities in mid-mantle geodynamics – affecting the stagnation of slabs and free upward motion of plumes.

Here, we consider two examples where non-linear phase boundary morphologies have been invoked to explain mid-mantle dynamics: (1) the intersection of the ringwoodite-to-bridgmanite plus periclase transition with the bridgmanite-to-akimotoite and ringwoodite-to-akimotoite plus periclase transitions, forming a 'branching' morphology, and (2) the curvature of the garnet-to-bridgmanite transition. Using simple mantle convection or circulation simulations, we find that the dynamic impact of these example phase transitions are limited by either the uniqueness of thermodynamic state or the low magnitude of the phase buoyancy parameter respectively. Therefore it is unlikely that these phase boundary morphologies will, by themselves, prevent material exchange across the mid-mantle.

## 1 Introduction

Jumps in mid-mantle material properties (including density) have been a feature of 1D seismological models since the 1940s (e.g., Jeffreys and Bullen, 1940; Dziewonski and Anderson, 1981; Kennett and Engdahl, 1991), and since the early 1950s have been associated with phase transitions (e.g., Birch, 1952). Since the equilibrium pressure of these transitions is dependent on temperature (via the Clapeyron slope $\gamma = \frac{dP}{dT}$), temperature anomalies in the mid-mantle can shift the depth of the transition, creating topography on the associated surface of the seismic discontinuity (e.g., Shearer and Masters, 1992). Where the Clapeyron slope is negative, this topography generates buoyancy forces that counteract thermal buoyancy (see Figure 1). Christensen and Yuen (1985) introduced a phase buoyancy parameter '$\mathbb{P}$' as a nondimensional number to describe the pro- or contra-convective strength of a phase change:

$$\mathbb{P} = \frac{\gamma \cdot \Delta\rho}{\alpha \cdot \rho^2 \cdot g \cdot D} \tag{1}$$

where $\alpha$ represents the thermal expansivity, $\rho$ and $\Delta\rho$ represent the characteristic density and density change due to the phase transition respectively, $g$ is the acceleration due to gravity, and $D$ denotes the thickness of the convective layer.





The density jump associated with the $Rw$ (Ringwoodite) $\rightarrow Brm$ (Bridgmanite) $+ Pc$ (Periclase) reaction can layer mantle convection in 2D (e.g., Christensen and Yuen, 1985) and 3D (e.g., Tackley et al., 1994; Bunge et al., 1997) numerical models if the phase buoyancy parameter is sufficiently negative. If the counter-convective effect of the phase transition is large, but not so large as to layer convection, global models enter a 'transitional regime' with impeded transfer across the phase boundary, and the deflection of downwellings and upwellings (e.g. Wolstencroft and Davies, 2011). In regional models mod-
erately counter-convective phase transitions, combined with high mid-mantle viscosity contrasts can result in the deformation of slabs as they enter the deep mantle (e.g., Čížková and Bina, 2019). The phase buoyancy parameter necessary to induce a layered or transitional mode of global convection varies as a function of the vigour of convection described by the Rayleigh Number given by

$$Ra = \frac{\alpha \rho g \Delta T D^3}{\kappa \eta} \tag{2}$$

where $\Delta T$ is the non-adiabatic temperature difference between the top and bottom of the convecting layer, $D$ is the thickness of the convecting layer, $\eta$ is the viscosity of the mantle, and $\kappa$ is the thermal diffusivity. At Earth-like vigour ($Ra \sim 8 \times 10^6$) Wolstencroft and Davies (2011) found a value for $\mathbb{P} < -0.21$ and $\mathbb{P} < -0.085$ for layered and transitional modes respectively.

The phase transitions $Ol$ (Olivine) $\rightarrow Wd$ (Wadsleyite) (associated with the 410 km depth discontinuity) and $Rw \rightarrow Brm + Pc$ (associated with the 660 km depth discontinuity, hereafter '660') are likely the most important in mid-mantle
dynamics (mineral physics suggests $\mathbb{P}_{\text{Ol-out}} = 0.024$ and $\mathbb{P}_{\text{Rw-out}} = -0.022$ – using parameters in Table 2). These, however, are not the only reactions in the mid-mantle; experimental and computational mineral physics has long shown that an extended series of phase transitions happen through the mid mantle, resulting in a complex patchwork of mineral stability fields (e.g., Stixrude and Lithgow-Bertelloni, 2011). Recently, seismologists have shown that the simple reactions do not fully explain the topography of the '410' and '660' seismic discontinuities (Cottaar and Deuss, 2016), and mineral physicists have proposed
geodynamic impacts of modelled and observed phase diagram morphologies (e.g., Chanyshev et al., 2022; Ishii et al., 2023). It is, therefore, interesting to consider these morphologies and to consider and in particular their impact on the Earth's mantle dynamics.

Recently, some numerical codes have begun to implement buoyancy forces using tables of computed thermodynamic properties (e.g., Dannberg et al., 2022; Li et al., 2024), and these forces (assuming spatial gridding in the tables and geodynamic
modelling doesn't 'miss' minor reactions) should all be 'complete'. The aim of this work is to consider from a simpler theoretical perspective how large a role these more complicated phase diagram morphologies could play and to inform intuitions about the extent that reactions described by a single Clapeyron slope approximate phase boundaries in Earth materials.

Here, we consider two examples – the post-spinel reactions via Akimotoite and the post-garnet reaction – which represent two basic morphologies beyond the well understood 'linear' phase boundary morphology: 'Branching' and 'Curving'
morphologies respectively.



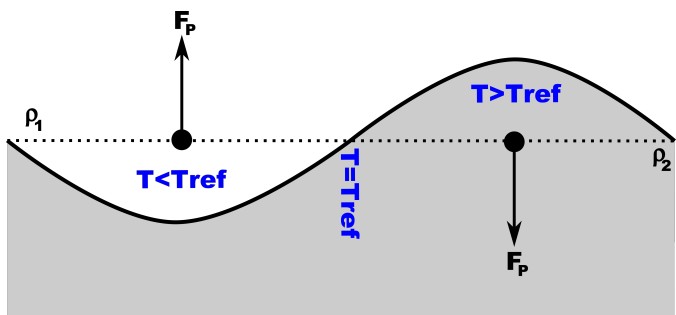

**Figure 1.** Generation of buoyancy forces due to thermally driven topography on the equilibrium depth of a phase transition with a negative Clapeyron slope. Temperatures are given relative to a reference temperature $T_{ref}$, defined as the temperature where the phase boundary sits at some reference depth (dotted line). Where $\gamma = \frac{dP}{dT} < 0$ and $T < T_{ref}$, the equilibrium pressure is greater than that at $T_{ref}$, so the phase boundary deepens. Since the less dense upper material ($\rho_1$) is laterally adjacent to denser material ($\rho_2 > \rho_1$), this produces a positive buoyancy force, opposing the effect of the thermal buoyancy.

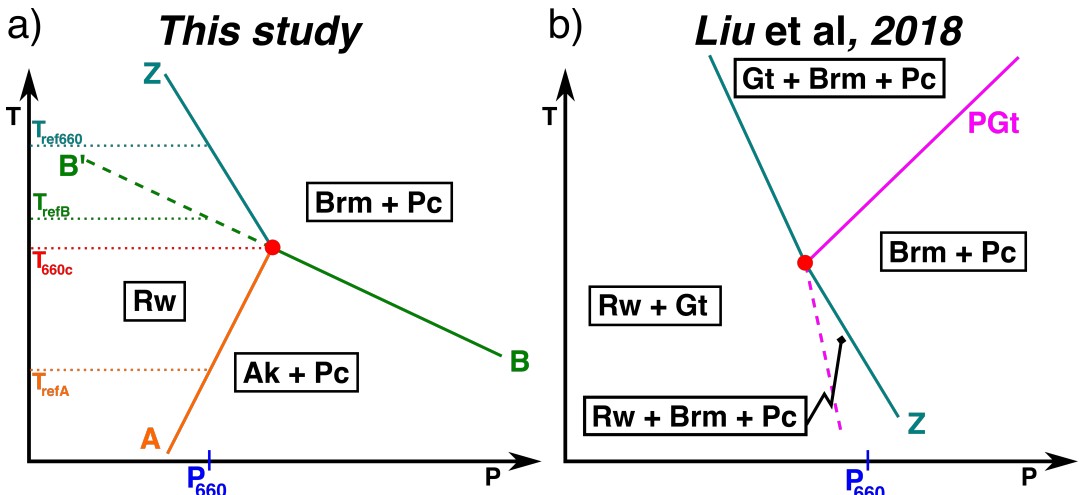

**Figure 2.** Sketch diagrams of branching phase diagram morphologies as implemented in geodynamic simulations. Temperature and pressure ranges are different between panes – the pressure at 660km depth ($P_{660}$) is included in for reference. *a)*: Sketch phase diagram of olivine system in the 660 phase boundary region. The reference temperature for reactions 'A' (orange line) and 'B' (green line) is calculated by projecting out from the triple point, described by temperature $T_{660c}$ – for reaction B this is done along the pseudo-boundary beyond the triple point (dashed line labelled 'B′′'). *b)*: Sketch pyrolite (60% Olivine, 40% Garnet) phase diagram implemented by Liu et al. (2018) at the intersection of the post spinel reaction (blue line labelled according to our convention as 'Z') and the post garnet reaction (magenta line labelled as 'PGt'). Note that the PGt implemented by Liu et al. in their geodynamic simulations is much shallower than the post garnet reaction we consider below since they are motivated by a different set of data (Hirose, 2002). Liu et al. (2018) do not consider the role of the post-garnet reaction at temperatures beneath the intersection – the dashed magenta line here represents this neglected necessary phase boundary.





## 1.1 Branching phase boundaries: post spinel reactions via akimotoite

At cooler temperatures, bridgmanite in the post-spinel reaction ($Rw \rightarrow Brm + Pc$, hereafter reaction 'Z') is replaced by the ilmenite group mineral Akimotoite (e.g., Yu et al., 2011) – $Rw \rightarrow Ak$ (Akimotoite) $+ Pc$ (hereafter reaction 'A') followed by transformation of $Ak \rightarrow Brm$ (hereafter reaction 'B') (See Figure 2 a). Recently, there has been interest in the potential

geodynamic role of reaction 'B' whose Clapeyron slope is also negative but significantly larger in magnitude than that of the global reaction 'Z', and so mineral physicists and seismologists have suggested that reaction 'B' might potentially provide a mechanism to aid the stagnation of slabs (e.g., Cottaar and Deuss, 2016; Chanyshev et al., 2022).

This is not the first contribution to the geodynamic literature that considers the effect of a 'branching' phase boundary morphology on mid-mantle dynamics. Using a phase function (after Christensen and Yuen, 1985), Liu et al. (2018) considered

the effect of a sharp post-spinel reaction and broad linear post-garnet reaction interacting on a plume, forming a similar branched morphology except flipped in temperature (i.e. the 'trunk' is on the cooler side, and the 'branches' are on the warmer side of the critical temperature – see Figure 2 b). The phase diagram used by Liu et al. (2018) does not conserve entropy and volume changes around the phase boundary intersection since beneath the intersection temperature they should still have a Garnet-out reaction, which they neglect to include. However, taking their implemented phase boundary buoyancy forces as-

is, they show that the post-garnet reaction with a positive Clapeyron slope has a strong impact on the local phase boundary topography and significantly reduces the counter-convective effect of the post-spinel reaction on the long lengthscale velocity field (Liu et al., 2018).

## 1.2 Curving phase boundaries: post garnet reaction with variable Clapeyron slopes

Recently, Ishii et al. (2023) suggested that due to a temperature-dependent heat capacity of Garnet at lower mantle temperatures,

the post-Garnet reaction would have a negative Clapeyron slope at cooler temperatures, and a positive Clapeyron slope at warmer temperatures (see Figure 3). Additionally, Ishii et al. (2023) suggest that due to the presence of a small amount of stishovite, the Post-Garnet reaction is univariant and sharp instead of a broad, divariant reaction, as has generally been assumed. Ishii et al. (2023) suggest that their temperature-dependent Clapeyron slope is responsible for slab stagnation whilst permitting plumes to pass unimpeded through the lowermost mantle transition zone. We describe this phase boundary morphology as a

'curving' morphology.

## 1.3 This contribution

In this contribution we illustrate the potential effects of these phase boundary morphologies. In Section 2 we describe the implementation of phase boundary topography buoyancy forces in TERRA, and how we modify the current implementation for a Curving and Branching morphology, and show the impact of these forces in simple thermal convection models to build

intuitions about the controls and limitations of their impacts on flow. In Section 3 we illustrate the (non)-impact of the Curving morphology on a mantle with a more Earth-like model setup.



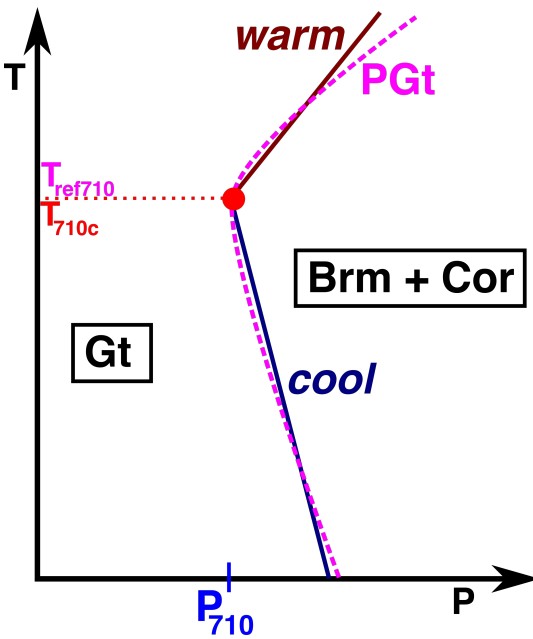

**Figure 3.** Sketch phase diagram of Post-Garnet reaction with temperature-dependent phase boundary. Smooth magenta dashed curve shows the phase boundary fitted by Ishii et al. (2023), straight navy and maroon lines sketch the implementation strategy here of treating the curved phase boundary as two straight boundaries meeting at a critical temperature. Pressure at 710 km depth ($P_{710}$) where the sheet mass anomaly is implemented is indicated. Phases Ishii et al. (2023) found to be stable indicated – $Cor$ = corundum.

## 2 Simple thermal convection models

To illustrate the potential dynamics that these diverse phase boundary morphologies can produce, we run simulations using the mantle convection code TERRA (e.g., Baumgardner, 1983; Panton et al., 2022) detailed in Table 1. We consider in turn the implementation of phase boundaries in TERRA, the implementation of branching phase boundaries and their possible effects, and finally the implementation and effects of curving phase boundaries.

### 2.1 Implementation of phase transitions

We implement all the reactions (post-Olivine, post-Spinel and the more complex examples considered below) using the sheet-mass anomaly method (e.g., Tackley et al., 1993; Bunge et al., 1997). This more simple method allows a direct consideration of phase boundary transition parameters. In a Boussinesq approximation, the method adds a buoyancy stress ($\Delta\sigma$) due to the topography of the phase transition as

$$\Delta\sigma = \gamma \cdot \frac{\Delta\rho}{\rho} \cdot (T - T_{\text{ref}}) \tag{3}$$

on the nodes at the reference depth of the phase transition. In this expression, $T$ is the temperature at the node and $T_{\text{ref}}$ is the reference temperature (see Figure 2, a) of the phase boundary (temperature where the phase boundary sits at the reference





| Simulation Number | $\gamma_{\text{cool}}$ [MPa.K$^{-1}$] | $\gamma_{\text{hot}}$ [MPa.K$^{-1}$] | $T_{710c}$ [K] | $\gamma_{660B}$ [MPa.K$^{-1}$] | $T_{660c}$ [K] | Regime |
|---|---|---|---|---|---|---|
| 100 | · | · | · | · | · | W |
| 101 | · | · | · | -4.4 | 1400 | W |
| 102 | 0 | 2.5 | 1200 | · | · | W |
| 103 | -1.5 | 2.5 | 1200 | · | · | W |
| 104 | -3 | 2.5 | 1200 | · | · | W |
| 105 | -9 | 2.5 | 1200 | · | · | W |
| 106 | -12 | 2.5 | 1200 | · | · | W |
| 107 | -18 | 2.5 | 1200 | · | · | S |
| 108 | 0 | 2.5 | 1850 | · | · | W |
| 109 | -1.5 | 2.5 | 1850 | · | · | W |
| 110 | -3 | 2.5 | 1850 | · | · | W |
| 111 | -9 | 2.5 | 1850 | · | · | W |
| 112 | -12 | 2.5 | 1850 | · | · | W |
| 113 | -18 | 2.5 | 1850 | · | · | S |
| 114 | 0 | 2.5 | 2500 | · | · | W |
| 115 | -1.5 | 2.5 | 2500 | · | · | W |
| 116 | -3 | 2.5 | 2500 | · | · | W |
| 117 | -9 | 2.5 | 2500 | · | · | W |
| 118 | -12 | 2.5 | 2500 | · | · | W |
| 119 | -18 | 2.5 | 2500 | · | · | S |

**Table 1.** Simple thermal convection simulations, varied parameters and dynamic regimes classified as – whole mantle convection ('W') or whole mantle convection with downwelling stagnation ('S').

depth (i.e. 410km or 660 km in depth)), $\Delta\rho$ is the density difference associated with the phase change and $\gamma$ is the Clapeyron slope of the phase transition.

## 2.2 Reference model

To compare the effect of including branching and curving morphologies, we first consider a reference model with a linear post-spinel reaction (Simulation Number 100). All the other simple convection models are based on this model setup. The
simulations are run in TERRA, using the parameters listed in table 2. The model is run for 4.5 Gyr (to quasi thermal steady state, see Figure A1), and the model is visualized in Figure 5a. We use an isoviscous rheology and take the Boussinesq approximation. In all simulations, as well as the deeper mantle transition zone reactions considered in this paper, we include the Olivine-out reaction at 410 km depth. We do not include the direct effect of the enthalpy of reaction on the temperature field. This effect is generally smaller than the effect of the phase boundary topography and has the opposite effect on the





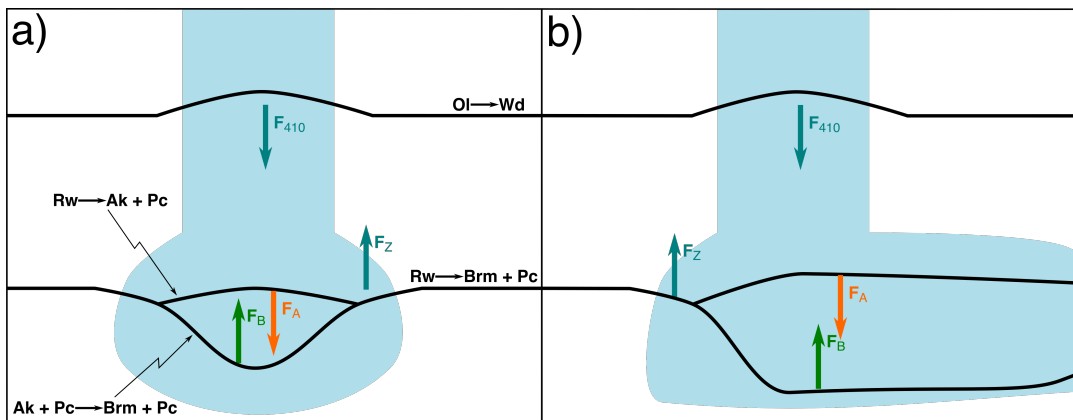

**Figure 4.** Sketches illustrating the forces due to phase boundary topography of reactions A & B in cool downwellings. *a)*: In a vertical slab, where the split phase transition region is comparatively small, and the opposing phase transition topography buoyancy forces act on near-identical regions; *b)*: And in a stagnating slab, where the phase transitions spread further apart the forces act on distinct regions

convection (e.g., Steinberger, 2007; Dannberg et al., 2022), hence the work here represents an upper bound on the potential effect. We do not include the effect of laterally varying composition on the relative strength of the reactions. Whilst the mantle as a whole has a pyrolytic composition ($\sim 60\,\%$ olivine and $\sim 40\,\%$ garnet in the upper mantle e.g. Bina and Helffrich 2013), compositional hetrogenities generated by melting at the surface result in regions enriched and depleted in garnet and pyroxene in the upper mantle – in entirely enriched regions (high in basalt) of the mantle, no post-spinel reaction is expected (and the post-garnet reaction – see below – would be at its maximum strength), and in entirely depleted regions of the mantle the post-spinel reaction should be at its maximum strength. We assume that each reaction affects all of the material around it, and thus provide an overstated upper bound on the influence of reactions on mantle flow.

### 2.3 Implementing branching phase transitions (post-spinel via akimotoite)

The sheet-mass anomaly approximation (Tackley et al., 1993) assumes that the length scale of the topography on the phase boundary is less than or similar to the length scale of the thermal structure that drives the flow. For a downwelling body whose temperature is 500 K below the critical temperature of the reactions 'A', 'B', and 'Z', and with Clapeyron slopes of $\gamma_A = +1.5$ and $\gamma_B = -6$ MPa/K, we estimate a maximum separation between the phase transition surfaces inside the downwelling body as on the order of 100 km. In our global convection models (with a comparatively high mantle viscosity of $10^{23}$ Pa s), our slabs end up much wider than this separation, allowing us to consider the phase transition boundary topography force from both the reactions 'A' and 'B' as acting at similar depths (see Figure 4). We are probably at the edge of where this approximation is valid (moving toward weaker and temperature dependent geological mantle rheologies would probably break this simplification for these reactions) but the theoretical understanding developed below suggests significant effects on global dynamics are unlikely. Given that this approximation overstates the dynamic effect branching reactions can have on mantle flows, we expect our



| Symbol | Parameter | Value | |
|--------|-----------|-------|---|
| $t$ | Simulation Duration | 4.5 Gyr | |
| $\rho_0$ | Reference Density | $4500 \, \mathrm{kg \, m^{-3}}$ | [a] |
| $\alpha$ | Thermal Expansivity | $2.5 \times 10^{-5} \, \mathrm{K^{-1}}$ | [b] |
| $k$ | Thermal Conductivity | $4 \, \mathrm{W m^{-1} K^{-1}}$ | [c] |
| $C_P$ | Specific Heat Capacity | $1100 \, \mathrm{J \, kg^{-1} K^{-1}}$ | [d] |
| $T_{\mathrm{surf}}$ | Surface Temperature | 300 K | |
| $T_{\mathrm{CMB}}$ | Core Mantle Boundary (CMB) Temperature | 3000 K | |
| $H$ | Internal Heating | $0 \, \mathrm{W \, kg^{-1}}$ | |
| $\gamma_{410}$ | $\frac{dP}{dT}$ of the $Ol \rightarrow Wd$ | $+1.5 \, \mathrm{MPa \, K^{-1}}$ | |
| $\gamma_{660}$ | $\frac{dP}{dT}$ of the $Rw \rightarrow Brm + Pc$ | $-1 \, \mathrm{MPa \, K^{-1}}$ | |
| $\eta_0$ | Reference Viscosity | $10^{23} \, \mathrm{Pa \, s}$ | [e] |
| $\Delta\rho_{410}/\overline{\rho_{410}}$ | Density change for $Ol \rightarrow Wd$ | 6.37 % | |
| $\Delta\rho_{660Z}/\overline{\rho_{660Z}}$ | Density change for $Rw \rightarrow Brm + Pc$ | 8.7 % | |
| $\Delta\rho_{660A}/\overline{\rho_{660A}}$ | Density change for $Rw \rightarrow Ak + Pc$ | 4.5 % | * [f] |
| $\Delta\rho_{660B}/\overline{\rho_{660B}}$ | Density change for $Ak + Pc \rightarrow Brm + Pc$ | 4.2 % | * [f] |
| $\Delta\rho_{\mathrm{PGt}}/\overline{\rho_{710}}$ | Density change for $Gt \rightarrow Brm + Cor$ | 3.13 % | † [g] |
| $\gamma_{\mathrm{warm}}$ | $\frac{dP}{dT}$ of the $Gt \rightarrow Brm + Cor$ on the warm arm | $2.5 \, \mathrm{MPa \, K^{-1}}$ | † [e] |

**Initial Condition:** All models initialized from a temperature field (identical across simulations) with small random anomalies away from the average temperature structure.

[a] – Reasonable typical mantle density, e.g. Dziewonski and Anderson (1981)

[b] – Typical values for mantle materials (p. 61, Porier, 2000)

[c] (p. 122, Clauser and Huenges, 1995)

[d] (Panton, 2020)

[e] (Deschamps and Cobden, 2022)

[f] (Kojitani et al., 2022)

[g] (Ishii et al., 2023)

* – for Ak simulation, not included in PGt or reference simulations

† – for PGt simulations, not included in Ak or reference simulations

**Table 2.** Values taken as constant across convection simulations.





|  | $\gamma_Z$ [MPa K$^{-1}$] | $\gamma_A$ [MPa K$^{-1}$] | $\gamma_B$ [MPa K$^{-1}$] | $T_{660c}$ [K] |
|---|---|---|---|---|
| Yu et al. 2011 | -2.9 | 1.2 | -6 | 1400 |
| Ye et al. 2014 | -2.5 | · | · | · |
| Hernández et al. 2015 | · | 1.65 | -3.5 | 1565 |
| Kojitani et al. 2016 | -1 | · | · | · |
| Kojitani et al. 2022 | · | 1.8 | -4.4 | 1440 |
| Chanyshev et al. 2022 | -0.1 | · | -8.1 | · |

**Table 3.** Values for parameters recently calculated by mineral physicists. '·' indicates that this value is not evaluated in the publication.

conclusions to be robust. Therefore, Equation 3 becomes

$$\Delta\sigma_{660} = \begin{cases} \gamma_Z \cdot \frac{\Delta\rho_Z}{\rho}(T_{660} - T_{\text{ref}660}) & T_{660} \geq T_{660c} \\ \\ \gamma_A \cdot \frac{\Delta\rho_A}{\rho} \cdot (T_{660} - T_{\text{ref}A}) + \gamma_B \cdot \frac{\Delta\rho_B}{\rho} \cdot (T_{660} - T_{\text{ref}B}) & T_{660} < T_{660c} \end{cases} \qquad (4)$$

where $T_{660c}$ is the temperature of the triple point, where reactions A, B, and Z meet (see Figure 2 a).

We do not have free choice over the values of the Clapeyron slopes and density changes at branching intersections, since entropy and volume changes are properties of state alone. The changes in entropy and volume from one state to another must be equal regardless of the path taken. With reference to Figure 2a, the entropy and volume changes must be the same whether the

assemblage changes from $Rw$ to $Brm + Pc$ via reaction 'Z' or via reactions 'A' and 'B'. Assuming $\Delta V \ll V$ and re-writing Equation 4 in terms of volume and entropy change we get

$$\Delta\sigma_{660} = \begin{cases} \frac{-\Delta S_Z}{V} \cdot (T_{660} - T_{\text{ref}}) & T_{660} \geq T_{660c} \\ \\ \frac{(-\Delta S_A - \Delta S_B)}{V} \cdot (T_{660} - T_{\text{ref}}) & T_{660} < T_{660c} \end{cases} \qquad (5)$$

As $\Delta S_A + \Delta S_B = \Delta S_Z$, the temperature-dependence of the phase boundary topography buoyancy force has to be the same either side of the triple point if we apply these forces using a sheet-mass anomaly approximation.

**2.4 Results for thermodynamically consistent branching phase transitions**

In order to demonstrate the (non-)effect of implementing a thermodynamically consistent branched morphology using the implementation described above, a simulation based on the reference case was run, but with $\gamma_A = 1.8\,\text{MPa K}^{-1}$ and $\gamma_B = -4.4\,\text{MPa K}^{-1}$ (i.e. the values of Kojitani et al. (2016, 2022), but these are similar to other published values for the Clapeyron slopes – see table 3). This is visualized after 4.5 Gyr in Figure 5b, where it is obvious that the simulations are very nearly





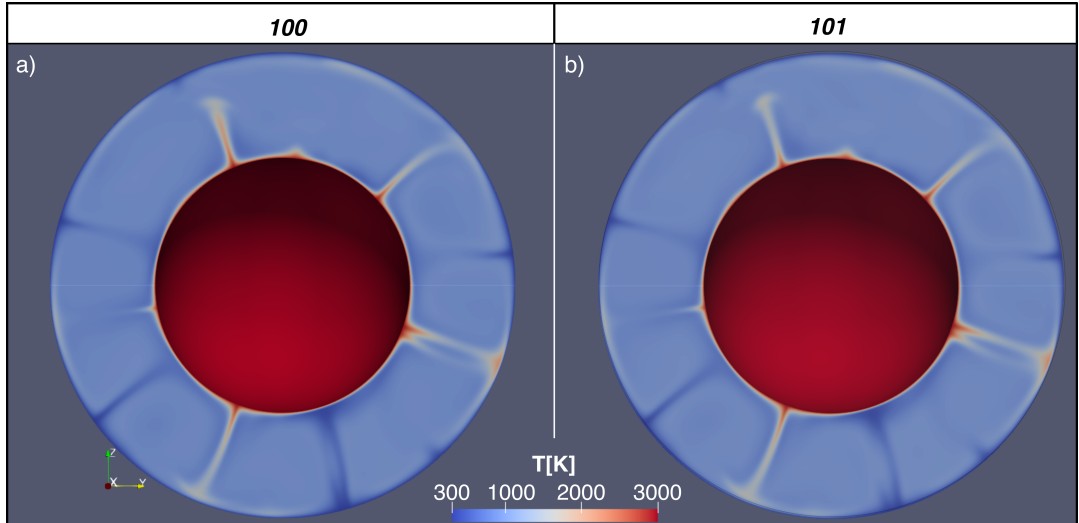

**Figure 5.** Visualisation of convection simulations after 4.5Gyr of evolution *a)* Reference case (Simulation Number 100) – post-olivine and post-spinel reactions included, but no other reactions included. *b)* Branching case (Simulation Number 101) – Akimotoite reactions implemented using values suggested by Kojitani et al. (2016, 2022). Note that there is no significant difference between the thermal structure of the simulations, as expected.

identical, highlighting that the combined phase boundary buoyancy forces of reactions 'A' and 'B' result in an extremely similar resultant force to what reaction 'Z' would have produced.

## 2.5 Implementing curving phase transitions (post-garnet)

As illustrated in Figure 3, the curving post-garnet reaction is here implemented using two straight lines to approximate the parabolic form suggested by Ishii et al. (2023). Since the phase boundary topography implied by Ishii et al. (2023) is relatively modest, the sheet-mass anomaly approximation in TERRA is suitable for application to this reaction. We calculate the buoyancy stresses at 710 km depth due to this topography as

$$
\Delta\sigma_{710} = \begin{cases} \gamma_{\text{warm}} \cdot \frac{\Delta\rho_{\text{PGt}}}{\rho}(T_{710} - T_{\text{ref710}}) & T \geq T_{710\text{c}} \\ \\ \gamma_{\text{cool}} \cdot \frac{\Delta\rho_{\text{PGt}}}{\rho} \cdot (T_{710} - T_{\text{ref710}}) & T < T_{710\text{c}} \end{cases} \tag{6}
$$

where $\gamma_{\text{warm}}$ and $\gamma_{\text{cool}}$ are the Clapeyron slopes above and below the critical temperature. By choosing the depth at which the post garnet phase boundary force is implemented at as 710 km (the minimum depth of the phase boundary found by Ishii et al. (2023)), this means that a single reference temperature $T_{\text{ref710}}$ describes both the warm and cool legs of the 'curved' reaction, and is also the temperature at which the Clapeyron slope of the reaction changes $T_{710\text{c}}$. To explore the role that this curving




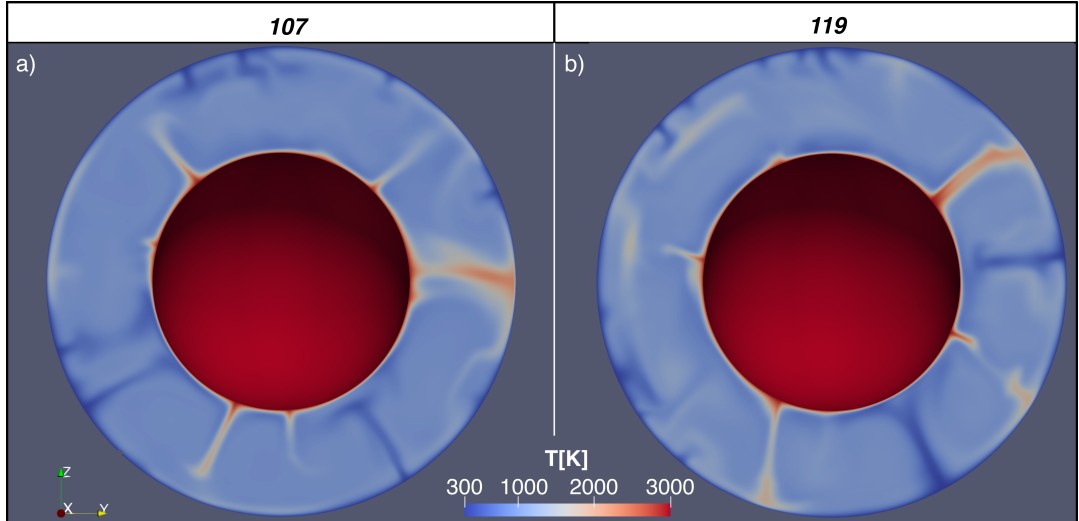

**Figure 6.** A great circle slice through the temperature field of 3D of convection simulations with PGt reaction present after $4.5$ Gyr, where $\gamma_{\mathrm{cool}} = -18\,\mathrm{MPa\,K^{-1}}$ after 4.5Gyr of evolution *a)* Simulation Number 107, $T_{710c} = 1200$K *b)* Simulation Number 119, $T_{710c} = 2500$K

reaction can apply we vary two parameters explicitly – $T_{710c}$ and $\gamma_{\mathrm{cool}}$ in the ranges of 1200 to 2500 K and 0 to $-18\,\mathrm{MPa\,K^{-1}}$ respectively over 18 simulations (see Table 1).

## 2.6 Results for curving (post-garnet) phase transition

Select simulations with a post-garnet inspired curving phase change are visualized in Figures 6 and 7. At the highest magnitude of $\gamma_{\mathrm{cool}}$ considered here ($-18\mathrm{MPa\,K^{-1}}$) (Figure 6a) many downwellings fail to enter the lower mantle. Decreasing the value of $\gamma_{\mathrm{cool}}$ (compare results in Figure 7a) results in a mode of convection similar to that seen for the reference case (see Figure 5a).

### 2.6.1 Effect of $T_{710c}$

As well as the magnitude and sign of the Clapeyron slope, another important parameter in determining the vigour of convection is $T_{710c}$ – which controls the portions of the mantle affected by pro- and contra-convective forcings from the phase boundary topography, with higher values of $T_{710c}$ meaning that more of the downwellings are effected by the cool-arm of the PGt reaction. With an increasing proportion of the Earth's mantle subject to a counter-convective forcing in cold downwellings in Figure 7b, downwellings penetrate the lower mantle less successfully to deeper depths. Considering Figure 6, where the

simulations have a higher magnitude of $\gamma_{\mathrm{cool}}$ ($-18$ vs $-12\,\mathrm{MPa\,K^{-1}}$) than in Figure 7 – we again see more downwellings stagnating in the right hand panel and large volumes of cold material building up in the mid mantle (e.g., stagnation at $\sim$ '8 O'Clock' figure 6b). In Figure 6a, by comparison, cold material is able to start entering the lower mantle with less negative thermal buoyancy (e.g., the downwelling at '9 O'Clock' descending with a bulge of stagnated material in the mid mantle, or at



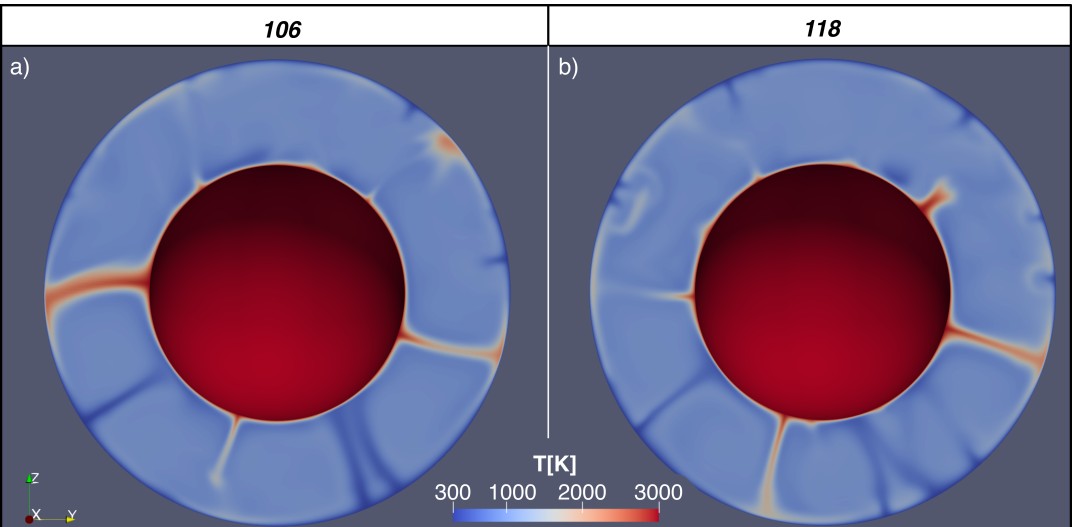

**Figure 7.** A great circle slice through the temperature field of 3D convection simulations with PGt reaction present after 4.5 Gyr, where $\gamma_{\mathrm{cool}} = -12\,\mathrm{MPa\,K^{-1}}$ after 4.5Gyr of evolution *a)* Simulation Number 106, $T_{710c} = 1200\mathrm{K}$ *b)* Simulation Number 118, $T_{710c} = 2500\mathrm{K}$

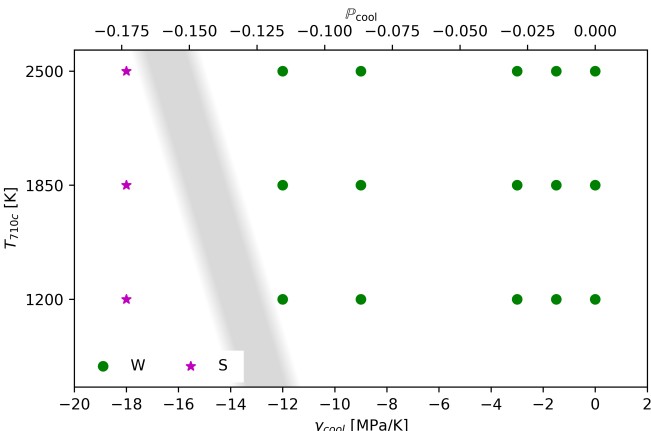

**Figure 8.** Regime diagram of model runs plotted on their values of $T_{710c}$ and $\gamma_{\mathrm{cool}}$. Simulations in the mode of whole mantle convection and in a mode with slab stagnation are marked 'W' and 'S' respectively. The rough region and inferred slope polarity of the boundary between these regimes is indicated by the grey shading.

'11 O'Clock a pile of cold material in the MTZ starting to descend into the lower mantle). This illustrates that increasing the value of $T_{710c}$, whilst not exerting a large enough influence to change the mode of convection here, is changing the magnitude of the local-counter convective effect on the downwellings.



### 2.6.2 Dynamic regimes

We categorize our 18 simulations into two groups based on the presence or absence of stagnant cold material in the mid-mantle; 'whole mantle convection' ('W') and 'whole mantle convection with downwelling stagnation' ('S'). Simulations that we categorize as 'S' have lower surface heat fluxes when they reach quasi-steady state that those in regime 'W' (compare magenta and green lines in Figure A1). The simulations with stagnating slabs and higher values of $T_{710c}$ (Simulation Numbers 113 and 119) have lower heat fluxes at 4.5 Gyr than the simulation with a lower value of $T_{710c}$ (Simulation Number 107 although this varies over the simulation duration). A regime diagram of these models plotted with their values of $T_{710c}$ and $\gamma_{cool}$ is shown in Figure 8. Note that there is no change of regime until much higher magnitude values of $\gamma_{cool}$ than those suggested by Ishii et al. (2023) to apply for the actual Earth. As illustrated in figures 7 and 6, there is some variation in downwelling behaviour within these regimes, but $T_{710c}$ plays a secondary role to $\gamma_{cool}$

### 2.7 Discussion on simple thermal convection models

Using these simple mantle convection models, we have been able to demonstrate the non-effect of a branching reaction (Figure 5) morphology of ringwoodite out via akimotoite and to demonstrate some of the behaviours possible for a curving phase boundary (Figures 7 and 6). By varying $T_{710c}$ and $\gamma_{cool}$ we have illustrated that the cold-arm of the post garnet reaction does not change the mode of global convection significantly until $\gamma_{cool}$ is an order of magnitude greater than that suggested from the experiments of Ishii et al. (2023). Similar to Bina and Liu (1995), we also found a dependence in convective style on the temperature where the Clapeyron slope changes, but found that $\gamma_{cool}$ had a much larger impact in our simulations than $T_{710c}$. Given that multiple factors about the set up of these simulations favour these phase boundaries being able to have a large effect on the mode of convection, this suggests that in the real Earth, this phase boundary morphology is unlikely to be responsible for the stagnation of slabs.

### 3 Thermochemical mantle circulation models at Earth-like vigour

The simple mantle convection models of the previous section are non-Earth-like in several respects. Firstly, we have assumed that there is no radiogenic or shear heating and that the models are incompressible – meaning that the simulated mid-mantles end up being much cooler on average ($\sim 950\,\mathrm{K}$) than the real Earth's mantle (e.g. $\sim 1650\,\mathrm{K}$ (Waszek et al., 2021)). Further, the convection simulations were isoviscous with a high reference viscosity, so did not represent the temperature and depth dependence of viscosity, which can influence downwellings at the base of the MTZ (e.g., Garel et al., 2014). Together, these result in a Rayleigh number of $\sim 10^5$ compared to the Earth's value of $\sim 10^7$ – which may understate the contra-convective effect since global phase transitions have a greater effect at higher Rayleigh number (e.g., Tackley et al., 1993; Wolstencroft and Davies, 2011). In the following sections we describe two significantly more sophisticated mantle circulation models incorporating ('TC1') and excluding ('TC0') the effect of the post-garnet reaction.





## 3.1 Method

Our more Earth-like simulations are also run in TERRA, but instead of a free-slip surface, the surface motion is imposed using
the plate motion history of Müller et al. (2022). These plate motions are scaled by a factor of 0.5 (that is, the plate motions
are slowed down to mantle velocities) so that the RMS surface motions match those produced by the simulated mantle in
convection (found during model initialisation, see below).

The model rheology is also more Earth-like than the previous simulations, with a reference viscosity closer to the Upper
Mantle viscosity (see Table 4). Viscosity varies according to

$$\eta = \eta_0 \cdot f_r \cdot e^{-E_a \cdot T'} \tag{7}$$

where $\eta$ is the local bulk viscosity, $\eta_0$ is the reference viscosity, $f_r$ is the radial viscosity factor (see Figure 9), $E_A$ is the
activation energy, and $T'$ is the non-dimensionalized temperature given by $T' = 0.5 - \frac{T - T_{\mathrm{CMB}}}{T_{\mathrm{surf}} - T_{\mathrm{CMB}}}$.

Composition is tracked on particles using the method of Stegman (2003). We model melting in the upper parts of the mantle,
generating enriched (basaltic) material at the surface, and depleted (harzburgitic) material in the source regions (Van Heck
et al., 2016). Variations in enrichment affect the intrinsic density – through most of the mantle this means that basalt is $\sim 2\,\%$
denser than lherzolite ('ambient mantle'), but between $660\,\mathrm{km}$ and $740\,\mathrm{km}$ depth, basalt is $5\,\%$ less dense than lherzolite as
the post-garnet reaction is deeper than the post-spinel, and basalt is enriched in garnet compared to lherzolite and harzburgite
(e.g., Yan et al., 2020). Aside from this 'basalt density filter' effect (which does not result in a strong concentration of basalt in
the deeper parts of the mantle transition zone in these simulations) we do not implement a compositional effect on the phase
boundary buoyancy forces.

### 3.1.1 Model initialisation

These simulations under go a two-stage initialization process. First, a convection simulation (with free-slip on both boundaries)
is run for 2 Gyr with no active chemistry, and without the Post-Garnet phase boundary forces implemented. The purpose of this
stage is to generate a quasi-steady state thermal structure. In the second stage of initialization, initial velocities of the Müller
et al. (2022) plate motion history are used for 400 Myr, particles are initiated, melting is turned on, and the post-garnet phase
boundary buoyancy forces are implemented into the simulation for TC1. This stage allows mantle structures to develop in
response to the imposed surface motions and also allows the effect of the additional phase boundary buoyancy force to come
into effect. Unrealistic flows and related effects arising from the onset of tectonic and igneous processes are allowed to dissipate
by the time the main simulation starts. The main simulation is then started with evolving scaled surface velocities imposed as
described above.

### 3.1.2 Choice of parameters for this comparison

We have chosen model parameters (see Table 4) to demonstrate the maximum plausible geodynamic effect of a sharp, curved
post-garnet reaction with the parameters suggested by Ishii et al. (2023). To do this we continue to assume that the phase tran-





| Symbol | Parameter | Value | |
|---|---|---|---|
| $\rho_0$ | Reference Density | $4500\,\mathrm{kg\,m^{-3}}$ | [a] |
| $\alpha$ | Thermal Expansivity | $2.5 \times 10^{-5}\,\mathrm{K^{-1}}$ | [b] |
| $k$ | Thermal Conductivity | $4\,\mathrm{W\,m^{-1}\,K^{-1}}$ | [c] |
| $C_P$ | Specific Heat Capacity | $1100\,\mathrm{J\,kg^{-1}\,K^{-1}}$ | [d] |
| $T_{\mathrm{surf}}$ | Surface Temperature | $300\,\mathrm{K}$ | |
| $T_{\mathrm{CMB}}$ | Core Mantle Boundary (CMB) Temperature | $4000\,\mathrm{K}$ | |
| $H$ | Internal Heating | $0\,\mathrm{W\,kg^{-1}}$ | |
| $\gamma_{410}$ | $\frac{dP}{dT}$ of the $Ol \rightarrow Wd$ | $+1.5\,\mathrm{MPa\,K^{-1}}$ | |
| $\gamma_{660}$ | $\frac{dP}{dT}$ of the $Rw \rightarrow Brm + Pc$ | $-1\,\mathrm{MPa\,K^{-1}}$ | |
| $\eta_0$ | Reference Viscosity | $4 \times 10^{21}\,\mathrm{Pa\,s}$ | |
| $E_A$ | Activation Energy | $1.75$ | |
| $\Delta\rho_{410}/\overline{\rho_{410}}$ | Density change for $Ol \rightarrow Wd$ | $6.37\,\%$ | |
| $\Delta\rho_{660Z}/\overline{\rho_{660Z}}$ | Density change for $Rw \rightarrow Brm + Pc$ | $8.7\,\%$ | |
| $\Delta\rho_{\mathrm{PGt}}/\overline{\rho_{710}}$ | Density change for $Gt \rightarrow Brm + Cor$ | $3.13\,\%$ | † [e] |
| $\gamma_{\mathrm{cool}}$ | $\frac{dP}{dT}$ of the $Gt \rightarrow Brm + Cor$ on the cool arm | $-1.5\,\mathrm{MPa\,K^{-1}}$ | † [e] |
| $\gamma_{\mathrm{warm}}$ | $\frac{dP}{dT}$ of the $Gt \rightarrow Brm + Cor$ on the warm arm | $2.5\,\mathrm{MPa\,K^{-1}}$ | † [e] |
| $B_{Bas}$ | Basalt Buoyancy Number | $0.44$ | |

**Initial Condition:** All models initialized from a temperature field (identical across simulations) with small random anomalies away from the average temperature structure.

[a] – Reasonable typical mantle density, e.g. Dziewonski and Anderson (1981)

[b] – Typical values for mantle materials (p. 61, Porier, 2000)

[c] (p. 122, Clauser and Huenges, 1995)

[d] (Panton, 2020)

[e] (Ishii et al., 2023)

† – for TC1, not included in TC0

**Table 4.** Values taken as constant across the mantle circulation simulations.





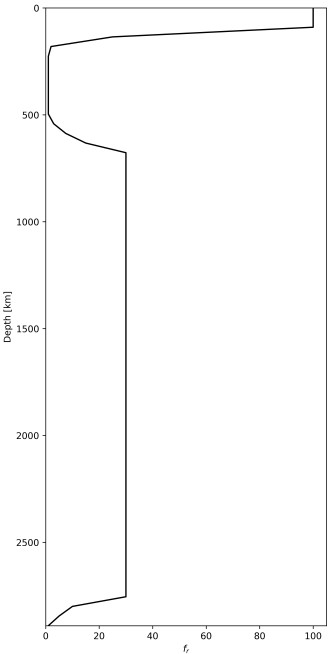

**Figure 9.** Radial viscosity factor ($f_r$) variation with depth. Note sharp steps in viscosity at the base of the lithosphere and upper mantle, as well as at the CMB

sitions affect all the mantle material at a node and use a comparatively high $T_{\text{CMB}}$ of $4000$ K to promote vigorous convection, a state in which counter-convective phase boundary topography forces are more impactful (e.g., Wolstencroft and Davies, 2011).

## 3.2 Results

The thermal and composition fields for the reference simulation ('TC0') and the MCM with the curved post-garnet phase boundary morphology ('TC1') are visualized in Figure 10 and Figure A2. In both simulations, cold slabs descend into the lower mantle, carrying enriched and depleted material to the CMB. Some slabs descend directly, while others stagnate in the mid mantle (e.g. the slabs beneath the North-West Pacific Figure 10 A & B about one or two O'Clock).

## 3.3 Discussion of thermochemical simulations

Simulations TC0 and TC1 appear to operate in a similar convectional regime and with comparable kinematics (Figure 10). There are differences between the two simulations – for example, the coldest regions associated with the Nazca slab (Figure 10 C & D, 7-10 O'Clock) appear more continuous in the simulation with PGt – however, the introduction of the novel phase boundary topography forces does not cause global slab stagnation, or create significant changes in regional slab stagnation. This suggests that while the new forces do have an effect on mantle processes, the effect is small and does not change the mode of mantle circulation.

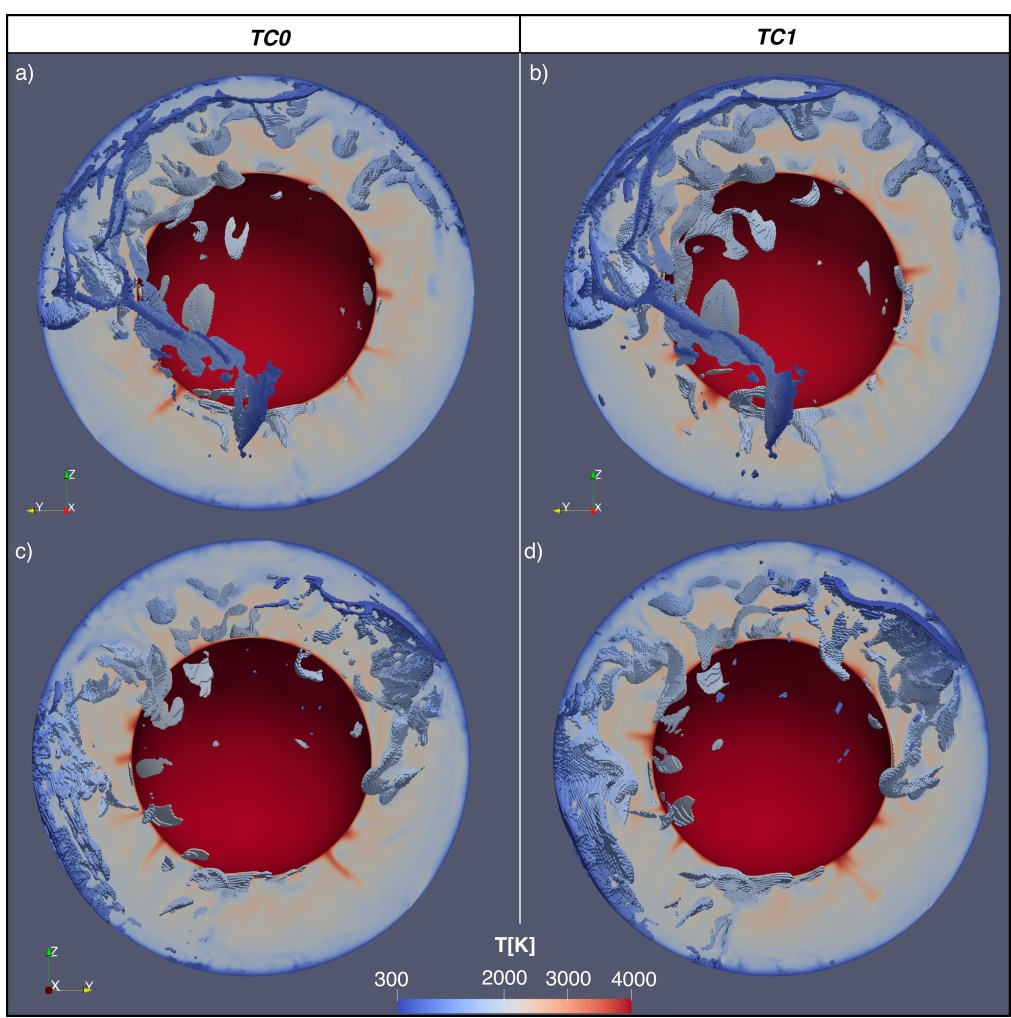

**Figure 10.** Comparison between reference MCM (TC0, *Left*) and MCM with curved PGt phase transition morphology (TC1, *Right*). Temperature visualized on latitudinal slice through 90°, as well as on 'voxels' beneath 500K below the radial average temperature. *a) and b)* show the Western Hemisphere; *c) and d)* show the Eastern Hemisphere. Composition visualized in Figure A2



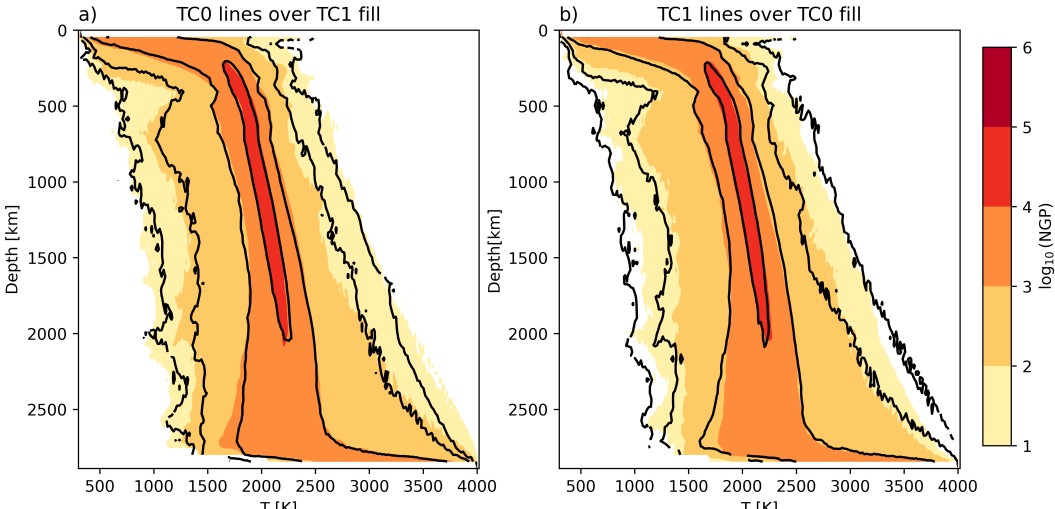

**Figure 11.** Radial temperature histograms coloured by number of grid points ('NGP') for PGt simulation TC1 and reference simulation TC0 with contours of the reference (TC0) and PGt (TC1) simulations' radial tempearture histograms overlaid respectively *a)* and *b)*.

The radial temperature histograms of the reference case, TC0, and simulation with the curved post-garnet transition, TC1, are extremely similar (Figure 11), suggestive of a near identical thermal evolution. In our previous convection simulations, models with large-scale slab stagnation had a distinct thermal evolution (Figure A1). When we look at the extreme nodes (the $10^2$ and $10^1$ Number of Grid Points (NGP) contours), we do see minor differences – for example, the coldest temperatures in Model TC1 are colder at most depths than for the reference model. These minor differences highlight that while there are buoyancy forces associated with a curved post-Garnet reaction, they do not significantly change the kinematics or thermal structure of the model.

## 4 Discussion

### 4.1 Branching phase boundaries: thermodynamically possible phase diagrams prohibit additional effect of post-spinel via akimotoite versus direct post-spinel

We have demonstrated above (Equation 5) that under the sheet-mass anomaly approximation, the phase boundary buoyancy forces for two branches must sum to the force of the 'trunk' reaction, and have shown that under this condition, no effect on mantle dynamics are observed (see Figure 5). This is potentially contrary to some of the suggestions in the mineral physics and seismology literature (e.g., Cottaar and Deuss, 2016; Chanyshev et al., 2022). Of course this prohibition only applies strictly where reactions 'A' and 'B' are close enough spatially that the forces can be treated as acting on the same region. As 'A' and 'B' move further apart a distinct counter-convective effect of reaction 'B' could emerge in extremely localised regions of slabs – these regions would be largest in slabs that are horizontal in the lowermost MTZ. Since these regions of slabs would sit within





a downwelling exerting a viscous downwards drag and still relatively close beneath reaction 'A' encouraging the descent of
material on to them, for reported values of $T_{660c}$, $\gamma_A$ and $\gamma_B$ it is difficult to envisage there being much of a dynamic effect in
the Earth's mantle.

### 4.2 Curving phase boundaries: phase buoyancy parameter ($\mathbb{P}$) implied by mineral physics too small to effect global dynamics

In our simple convection simulations, we were able to produce downwellings that stalled in the mid-mantle whilst upwellings
rose unimpeded through the post-garnet transition at values of $\gamma_{710cool}$ much greater than those suggested for PGt. Furthermore,
in a simulation with realistic surface motions and a curved phase boundary with parameters similar to those suggested for the
post-garnet reaction by Ishii et al. (2023) we observed no additional slab stagnation compared to a simulation without the
curved phase boundary. This result is expected, as the phase buoyancy parameter (using the parameters in Table 4) for the cool
post garnet is $-0.014$ – much less than that which would be expected to induce a layered or transitional dynamic regime. This
is a much smaller magnitude than the value of $-0.10$ found by Ishii et al. possibly due to them assuming the thickness of the
convecting layer is the above the phase boundary, whereas we assume it is the full depth of the Mantle. This is the choice used
elsewhere in the geodynamic literature (e.g. Tackley et al., 1994; Bunge et al., 1997; Wolstencroft and Davies, 2011), since this
is the thickness of the layer that would convect without the presence of an extremely counter-convective phase transition.

It is worth considering how well current theoretical models of a global linear reactions align with the temperature-restricted
reactions considered in our simulations.

For example, at a Rayleigh number of $1.4 \times 10^5$ (the same as in our simulations from Section 2), Wolstencroft and Davies
(2011) predicted regime changes from whole-mantle convection to 'transitional' convection at $\mathbb{P} < -0.25$ and to layered con-
vection at $\mathbb{P} < -0.32$. These thresholds are considerably more negative than the value of $\mathbb{P} \sim -0.15$ at which we see a regime
change (see Figure 8). This discrepancy may be explained by the temperature-restriction, which could truncate cold thermal
structures, making them more vulnerable to downwelling stagnation (Tackley et al., 1993). Wolstencroft and Davies (2011)
assume their regime boundaries follow the form $\mathbb{P} = \alpha Ra^{-\beta}$, where $\alpha$ and $\beta$ are constants. As the Rayleigh number increases,
the slopes of these curves tend to flatten. This suggests that the disparity in the value of $\mathbb{P}$ at which regime changes occur for
global and regional reactions may be less pronounced at high Rayleigh numbers than in the simpler convection simulations
presented here.

## 5 Conclusions

Using a combination of simple mantle convection models and selectively considering more complex mantle circulation models
we have illustrated some of the dynamic effects that non-linear phase boundary morphologies can produce when applied to
geodynamic simulations. Where entropy and volume changes for a single reaction are dependent on temperature, as Ishii et al.
(2023) have proposed for the post-garnet reaction, then distinct counter-convective and pro-convective forcings are possible on



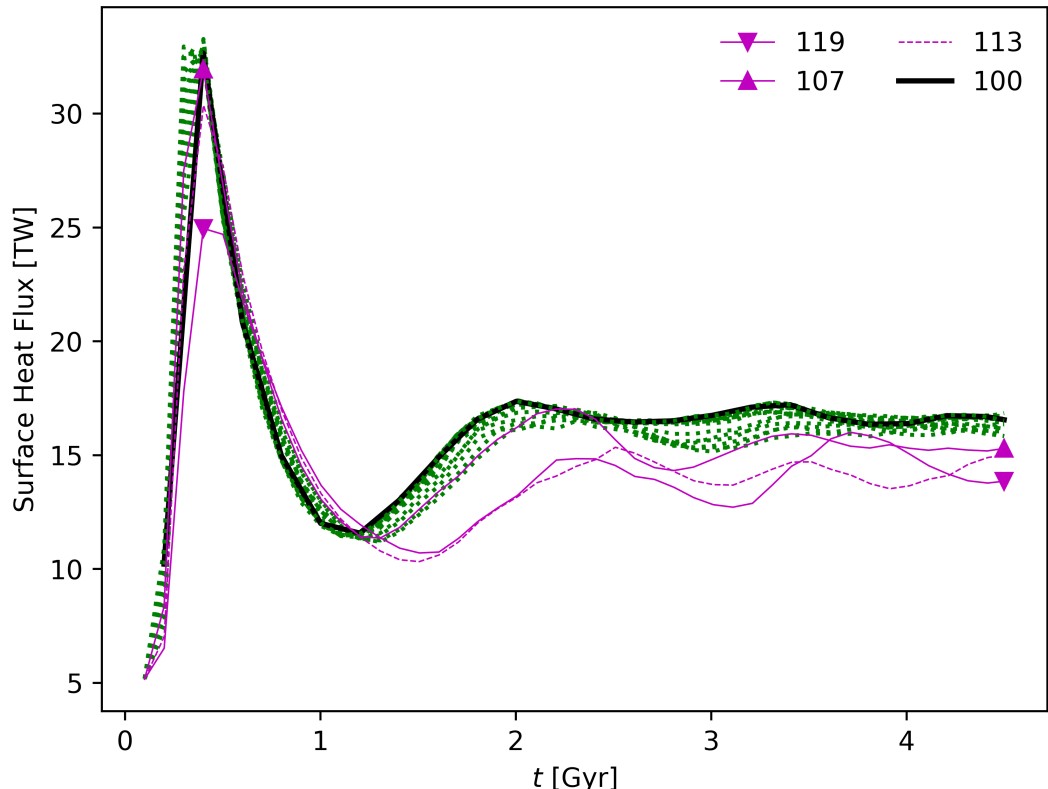

**Figure A1.** Evolution of surface heat flux of simple convection simulations in Table 1. Simulations are coloured according to their regime of convection in Figure 8, with select simulations highlighted in legend – Simulation Number 100 in heavy black line is the reference case, and Simulation Numbers 107, 113 and 119 are in magenta with downwards pointing triangle, dashed line, and upwards pointing triangle markers respectively are all in regime 'S'. Other simulations (in regime 'W') are in green dotted line – see discussion of dynamic regimes in Section 2.6.2.

upwellings and downwellings – but for the density changes and Clapeyron slopes suggested for the post-garnet reactions, this does not significantly affect global dynamics.

We have also highlighted that changes in entropy and volume must be conserved where reactions branch – and this is a critical constraint on the phase boundary buoyancy forces; the post spinel reaction remains a good approximation for total geodynamic effect of the olivine system reactions at the base of the Mantle Transition Zone.

**Appendix A**





**Figure A2.** Comparison between reference MCM (TC0, *Left*) and MCM with curved PGt phase transition morphology (TC1, *Right*). C-Value, describing composition between depleted (0, Harzburgitic), ambient (0.2, Lherzolitic), and enriched (1, Basaltic) visualized on latitudinal slice through $90°$, as well as on 'voxels' beneath 500K below the radial average temperature.*A* and *B* show the Western Hemisphere; *C* and *D* show the Eastern Hemisphere.



*Author contributions.* GTM: Conceptualisation, Methodology, Software, Investigation, Visualisation, Writing – original draft preparation

*JHD*: Conceptualisation, Methodology, Supervision, Funding Acquisition, Resources, Writing – review & editing

*RM*: Supervision, Methodology, Writing – review & editing

*JP*: Supervision, Methodology, Writing – review & editing

*Competing interests.* The authors declare that they have no conflict of interest.

*Acknowledgements.* This work used the ARCHER2 UK National Supercomputing Service (https://www.archer2.ac.uk) for all simulations and visualisations. GTM is funded by the College of Physical Sciences and Engineering, Cardiff University. JHD, RM and JP are funded by the NERC large grant "MC$^2$ - Mantle Circulation Constrained" (grant codes NE/T012633/1 and NE/T012595/1).





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
