# Peer review of "On the global geodynamic consequences of different phase boundary morphologies"

_EGUsphere, 2024_

## Author Comment (AC1)

Dear Anonymous reviewer,

Many thanks for your review. We have responded to each of the comments you made in detail below, and have revised the text as indicated. Line numbers indicate lines in the 'tracked changes' pdf that we will submit to the editor today.

Kind Regards,

Gwynfor Morgan & co-authors.

| RC1.01 | This manuscript presents a compelling exploration of the effects of branching and curved phase transitions on the stagnation of downgoing plates or cold anomalies in the upper and lower mantle. The study investigates the hypothesis that these transitions influence the likelihood of slab stagnation at various depths. The findings suggest that while these transitions may exert stagnation-supporting forces, the magnitude of these forces is insufficient to produce a discernible impact in global convection models. As such, the manuscript qualifies as a null-result paper—a less common but nonetheless important contribution to the field. |
|---|---|
| | Thank you. |
| RC1.02 | The manuscript is well-written, and the results are presented in a clear and logical manner. However, some figures could benefit from refinement (detailed suggestions are provided below). The structure of the paper is somewhat unconventional, with Sections 2 and 3 each resembling standalone studies, while the discussion and conclusions synthesise findings from both sections. |
| | Thanks, we hope the figures are clearer in the revised manuscript |
| RC1.03 | My primary concern is the limited motivation provided for conducting and publishing this study. While the authors cite two references that propose the 'branching' mechanism (Cottaar and Deuss, 2016; Chanyshev et al., 2022) and one for the 'curving' mechanism (Ishii et al., 2023) as contributors to slab stagnation, the rationale for exploring these mechanisms further is not sufficiently emphasised. I would encourage the authors to elaborate on why these mechanisms are worth investigating and, even if they are shown to have minimal relevance for Earth-like conditions in their models, to identify the conditions under which they might play a more significant role. The authors briefly address this for the 'branching' mechanism, suggesting relevance for stagnating flat slabs, but do not provide a similar discussion for the 'curving' mechanism. At this stage, it is unclear whether addressing this issue would require additional experiments (which would constitute a major revision) or could be achieved using existing results (a minor revision). |
| | Thank you for your comment. We did briefly motivate our study in the introduction (lines 44 – 56 in the revised manuscript) and this included a comment about the dynamic interest in the curved post-garnet (line 83-84 in the revised manuscript – again this is being invoked to explain slab stagnation in the mid-mantle). Whilst our response to peer review was being prepared similar arguments relating to post-spinel were discussed in Dong, |

| | |
|---|---|
| 2025 – we have added a passing reference to this (ln 84 -86 in revised manuscript, and also consider it in our concluding discussion (ln 363-ln 374) ).
We have added some text related to the conditions under which these phase boundary morphologies could play a more significant role (line 358-362 in the revised manuscript).
Dong, J., Fischer, R.A., Stixrude, L.P. *et al*. Nonlinearity of the post-spinel transition and its expression in slabs and plumes worldwide. *Nat Commun* **16**, 1039 (2025). https://doi.org/10.1038/s41467-025-56231-z | |
| RC1.04 | **Specific Comments**
   1. **Lines 120–123:**
"For a downgoing body whose temperature is 500 K below the critical temperature of the reactions 'A', 'B', and 'Z', and with Clapeyron slopes of $\gamma_A = +1.5$ and $\gamma_B = -6$ MPa/K, we estimate a maximum separation between the phase transition surfaces inside the downgoing body as on the order of 100 km".

Could you include this calculation, perhaps in the supplementary materials? |
| | We have edited the text to include the assumed pressure gradient and the calculation in the mid-mantle on line 151 in the revised manuscript – the calculation becomes straightforward but we have included it inline. |
| RC1.05 |    1. **Line 198:**
Given the limitations of the models discussed in Section 3, it might be helpful to note at the beginning of Section 2 that the initial set of models is not intended to be Earth-realistic. A cross-reference to the explanation in Section 3 would be beneficial for readers who may skim certain sections. |
| | We have amended the text to provide this sign-posting. (ln 101-103 in revised manuscript) |
| | **Figure-Specific Comments** |
| RC1.06 |    1. **Figure 1:**

     Consider marking the upper material (density $\rho_1$) with a colour to make its presence more apparent. Additionally, reposition the text boxes for $\rho_1$ and $\rho_2$ to clearly associate them with the bulk material, avoiding any potential confusion with density variations along the dotted line. |
| | These edits have been made to figure 1. |
| RC1.07 |    2. **Figures 2 and 3:**

     These figures effectively summarise the phenomena under investigation. You might consider merging them into a single figure with two or three panels for better visual coherence. |
| | We wish to keep figures 2 &3 separate to help distinguish between the 'branching' post-garnet reaction modelled by Liu *et al* (2018) and the 'curving' post-garnet we consider here. We are happy to be directed by the editor if this is counter to the style of *Solid Earth*. |

| | |
|---|---|
| RC1.08 | 3. **Figures 5, 6, and 7:**
    ○ Merge these figures into a single composite figure with six panels. Labelling each panel (e.g., with text in the red centre or a corner) would facilitate direct comparison between simulations, particularly since the text frequently refers to differences between Figures 6 and 7.
    ○ To address the local versus global nature of stagnation phenomena, consider adding supplementary material showing the 3D variations of your results. Options include additional slices, volume elements (similar to Figure 10), or a video of a rotating cross-section. |
| | In the revised version of the manuscript, we have merged figures 6 & 7 – the combined figure is numbered figure 6 in the revised manuscript. Figure 5 relates to the akimotoite simulation and a reference, so we choose to leave it as a separate figure to keep the suites of simulations clearly distinct. To improve the discrimination between dynamic regimes, we introduce a discussion of mass flux and radial velocity rms (figure A3, and lines 209 and 227-240) in the revised manuscript) as part of our response to Scott King's review. |
| RC1.09 | 4. **Figure 8:**

The diagonal line separating the two regimes does not appear to be well-supported by the data. A more accurate representation might involve marking the region between $\gamma\_cool$ = -17 and -13 as a transitional zone for any $T\_710c$ value. Please clarify in the text how you inferred the slope of the line and why it is presented as such. |
| | This figure is now numbered 7. We have provided some additional explanation in the text of how we picked the slope of the regime boundary, including a reference to the mass flux and radial velocity rms. The other reviewer is correct – that the boundary of the regimes in $\gamma\_cool$ has some dependence on $T\_710C$ is not really the key point of the figure. |
| RC1.10 | 5. **Figure 9:**
• Increase the font size for axes and labels.
• Provide a rationale for the behaviour of the radial viscosity factor, presumably designed to replicate Earth's mantle structure.
• Clarify the factor's values at the bottom of the lower mantle and in the upper mantle, as the graph suggests these may approach zero, which would imply $\eta = 0$ according to Equation 7. |
| | This figure is now numbered 8. We have amended this figure so $f\_r$ is plotted on a $\log\_{10}$ scale to emphasise $f\_r$ drops to 1, not to near 0, as well as some explanation for why we have chosen this radial viscosity gradient. |
| | **Minor Typographical Errors** |
| RC1.11 | 1. Line 46: "consider these morphologies and to consider and in particular" – likely an extra "and" |
| | (line 47 in the revised manuscript) Resolved. Thanks for indicating these errors to us. |
| RC1.12 | 2. Line 105: "the model is run" |

| | |
|---|---|
| | We have attempted to clarify the language here (line 117 in the revised manuscript) |
| RC1.13 | 3. Line 199: "the simulated mid-mantles" |
| | We have attempted to clarify the language here (line 253-254 in the revised manuscript) |
| RC1.14 | 4. Line 281: "the full depth of the Mantle" – unnecessary capitalisation of "Mantle" |
| | Resolved. Thanks for indicating these errors to us. |
| | I hope these suggestions help refine and strengthen your manuscript. |

---

## Author Comment (AC2)

Dear Scott King,

Many thanks for your review. We have responded to each of the comments you made in detail below, and have revised the text as indicated. Line numbers indicate lines in the 'tracked changes' pdf that we will submit to the editor today.

Kind Regards,

Gwynfor Morgan & co-authors.

| RC2.01 | This is an interesting and generally well-written contribution. There is a lot of discussion about phase transformations so this work is timely. I noted the presentation as "good" because I struggled with some of the figures and felt the author's need to do better. The text is clear and concise—almost terse—but fully understandable. With some revision to the figures the presentation would be excellent. |
|---|---|
| | Noted – we hope that the amendments have made the text less terse and easier to understand. |
| RC2.02 | I struggle because the authors neglect temperature-dependent rheology until the last section, especially because we have shown that it is important in layering 20+ years ago (King and Ita, 1995) but, I recognize that the authors are trying to do a clean series of calculations and keep things as simple as possible. At the very least, they need to acknowledge the important role temperature-dependent rheology can play. |
| | We have added this discussion into the revised manuscript (lines 94-96). We hope the additional text clarifies the limitations of our rheology |
| RC2.03 | The work uses the Bousinessq approximation and does not include the latent heat associated with the phase changes. That is the correct approximation (latent heat has the same non-dimensional terms as adiabatic compression and shear heating, so if you include it you should at least use extended Bousinessq). My question is have you thought about the role of latent heat? Is it secondary? You should alert the reader to this upfront. An old but useful ? reference might be Ita and King, 1994 where we found the formulation of the equations wasn't a major factor suggesting that is right. I don't know whether this 30 year old work stands the test of time or not. |
| | We did briefly discuss this – in our discussion of the reference simulation (line 122 in the revised manuscript). We have added an additional citation to the Ita & King 1994 paper and have modified our language slightly to clarify our meaning. |
| RC2.04 | Line 37: This is odd coming from me—because I am a big fan of non-dimensional formulations—but it would help you communicate to the non-geodynamics deep earth reader if you listed the related Clapeyron slopes along with values of P here. |
| | We have listed these in the revised manuscript (line 38 of the revised manuscript). |
| RC2.05 | Line 40: You should list Table 1 before listing Table 2 (or reverse the order) |

| | |
|---|---|
| | This has been corrected. Thanks for noting this error. |
| RC2.06 | Line 48: Actually, Ita and King, 1994 and 1998 did what you describe in the much more distance past… at least for the olivine system reactions, there wasn't enough pyroxene/garnet data to do that part. |
| | Citation to Ita & King 1994 added, and the sentence modified slightly, apologies for the insufficient reference to literature. |
| RC2.07 | Lines 54: I'm not sure Branching or Curving should be capitalized… that's a copy editor thing. |
| RC2.08 | Line 78: Post-Garnet -> post-garnet |
| RC2.09 | Lines 84-85: Curving -> curving; Branching -> branching |
| RC2.10 | Figure 3: Post-Garnet -> post-garnet |
| | Thanks for indicating these errors to us. Corrected. |
| RC2.11 | Line 106: Because you are focused on slabs/downwellings, this seems to be a real short coming. You should call attention to it for the reader. Here I think about Christensen, 1984 "In almost all cases power-law rheology leads to considerably different flow patterns and heat transfer properties than those predicted for Newtonian convection." |
| | We have added some discussion on this to this section (lines 126-130 in revised manuscript). Thanks |
| RC2.12 | Line 108: Olivine-out -> olivine-out |
| | Corrected, thank you. |
| RC2.13 | Figure 5, lines 144-146: presenting two slices through a 3D model is not very intuitive as we know (e.g., Tackley et al., 1993) that different behavior can be happening in different parts of the sphere. I was going to suggest showing radial correlation functions or maps at the 660 km depth but, I saw later that you present radial temperature histograms (Fig. 11) and so you must have the code to do this. I would find that more persuasive. I realize the challenge is that when you have some slabs stagnating and some not this might be misleading but, I think those would be more reliable than the single slices. |
| | Temperature histogram comparing simulations 100 and 101 now included as figure A1 in revised manuscript. |
| RC2.14 | Lines 160-164: I wonder if you can come up with a way to quantify stagnation or not a bit better. I think we used to use things like the reduction in radial velocity near the transformation. I admit it might suffer the challenges I brought up regarding radial correlation functions but, I find the reliance on patterns—especially when not shown as in this passage—to be unsatisfying. |
| | End-of simulation radial RMS velocity and mass flux at 720 km depth added as figure A3 and discussed in Sections 2.6.1 (line 209-213 and 2.6.2 (line 227-240 in revised manuscript). |
| RC2.15 | Line 169-168: This wording is a bit cumbersome. Maybe something more like, "As we increase the proportion of the donwelling subject to counter-convective forcing, stagnation becomes more likely. |
| | We have clarified the language here (line 202 in revised manuscript). |
| RC2.16 | Figures 6 and 7: Here I found the images and the text unsatisfying. I believe you would make a stronger case if you had a more quantitative measure. It |

| | took a lot of flipping back and forth to try and see the difference between these. |
|---|---|
| | We have added radial RMS velocity and mass flux at 720 km depth plots to our appendix (figure A3) and discussed them in the main text (see response to RC2.14). |
| RC2.17 | Figure 8: The slope of the grey band (not rough region) is not constrained by the calculations. I assume the authors are using theory to guide the slope. It is unfortunate that they have so many calculations with Pcool greater than -0.025 but, I would not suggest they leave any off. It appears that those calculations could have been used to better determine the line (ahh, hindsight is 20/20). Plotting the change of regime suggested by Ishii et al. (2023) would help to make the point in lines 184-185. My impression is that the slope is not the important point of the figure, the point is that the change happens well short of where Ishii et al. predict. |
| | This figure is now figure 7. We have added an indication of the Ishii et al data for the PGt reaction to this figure. We have changed the sign of the slope since inspecting the mass flux data and the radial velocity data suggests simulation 107 is close to whole-mantle convection, but we still have some stagnating downwellings in the temperature field, suggesting 107 is close to the transition between the two regimes. You are correct, the main point of the figure is that the change in dynamic regime happens at much higher phase buoyancy parameter than Ishii et al suggest. We note this in the main text on lines 220 and 225 in the revised manuscript. |
| RC2.18 | Lines 212-216: There is a problem with the Frank-Kamenetskii rheology (equation 7) when used for slabs. Because of the exponential it is too weak (see Javaheri et al, 2024 or King, 2009). It has been shown some time ago that rheology matters (King and Ita 2005). |
| | The text now includes some discussion about the possibility that increasing the temperature-dependence of the viscosity will make the downwellings less likely to stagnate (line 270-274 in revised manuscript). This possibility doesn't change the conclusions about the role of the garnet-out reaction on Earth's mantle dynamics. |
| RC2.19 | Section 3.2 I find this section to be more persuasive because of Figure 11. Adding one more calculation without the temperature-dependent rheology would be useful and would mitigate some of my concerns over rheology above. |
| | We do not feel that this further calculation is necessary – we have shown that for the simplest possible case PGt with the currently proposed value of the phase buoyancy parameter is an order of magnitude away from having a significant geodynamic effect. In the revised text we explain how the rheological simplifications of a 'weak' temperature dependency we apply in our circulation model move us towards conditions where downwelling stagnation is more likely, not less. For the conclusion we reach we don't feel that further calculations are necessary to apply the understanding to the Earth. This is discussed on lines 273-274 and 317-320 of the revised manuscript |

| RC2.20 | References: |
| --- | --- |
| | Ita, J. J. and S. D. King, The influence of thermodynamic formulation on simulations of subduction zone geometry and history, *Geophysical Research Letters*, 125, 1463-1466, 1998. |
| | Ita, J. J., and S. D. King, The sensitivity of convection with an endothermic phase change to the form of governing equations, initial conditions, aspect ratio, and equation of state, *Journal of Geophysical Research*, 99, 15,919-15,938, 1994. |
| | King, S. D., On topography and geoid from 2D stagnant-lid convection calculations, *Geochemistry, Geophysics, Geosystems,* 10, Q3002, 2009.  doi:10.1029/2008GC002250 |
| | King, S. D., and J. J. Ita, The effect of slab rheology on mass transport across a phase transition boundary, *Journal of Geophysical Research*, 100, 20,211-20,222, 1995. |
| | Pejvak Javaheri, Julian P. Lowman, Paul J. Tackley, 2024. Spherical geometry convection in a fluid with an Arrhenius thermal viscosity dependence: The impact of core size and surface temperature on the scaling of stagnant-lid thickness and internal temperature, Physics of the Earth and Planetary Interiors, 349, 107157, https://doi.org/10.1016/j.pepi.2024.107157. |
| | Tackley, P.J., Stevenson, D.J., Glatzmaier, G.A. and Schubert, G., 1993. Effects of an endothermic phase transition at 670 km depth in a spherical model of convection in the Earth's mantle. *Nature*, *361*(6414), pp.699-704. |